# Effects of Alkaline Mineral Complex Supplementation on Growth Performance, Meat Quality, Serum Biochemical Parameters, and Digestive Function of Fattening Lambs

**DOI:** 10.3390/ani16010106

**Published:** 2025-12-30

**Authors:** Qing Mu, Jiawei Ai, Zhiqiang Gao, Shujun Tian, Xiaoyong Chen

**Affiliations:** College of Animal Science and Technology, Hebei Agricultural University, Baoding 071000, China; muqingisme@163.com (Q.M.); ajwchn@163.com (J.A.); gaozhiqiang200614@163.com (Z.G.)

**Keywords:** alkaline mineral complex, antioxidant capacity, growth performance, meat quality, rumen microbiota

## Abstract

In intensive animal production, lambs are commonly fed high-concentrate diets to achieve rapid growth. However, this practice often induces metabolic disturbances, including oxidative stress and digestive disorders, ultimately compromising growth performance and meat quality. Our study investigated the effects of alkaline mineral complex (AMC) supplementation on the growth performance, meat quality, serum biochemical parameters, and digestive function of fattening lambs. The results demonstrated that supplementation with AMC at 3 g/d per lamb improved average daily gain (ADG) and feed efficiency. It also enhanced meat quality by increasing the meat redness (a*) and intramuscular fat (IMF) content. These benefits appear to be linked to an improved antioxidant status and positive changes in the rumen environment. In conclusion, AMC can be used as an effective feed additive to support the healthy growth and meat quality of lambs raised on high-concentrate diets.

## 1. Introduction

In recent years, lamb meat has become one of the most important meat products globally, with a steadily increasing demand, which promotes the rapid development of the lamb industry [1]. As living standards improve, consumer preferences are shifting from basic nutritional needs toward high-quality and health-oriented foods, leading to more stringent and refined requirements for meat quality [2].

In many major agricultural regions worldwide, intensive housing systems have become the dominant mode of lamb production. Compared to traditional grazing, high-concentrate diets are widely used under confined conditions to supply energy and protein rapidly, promoting accelerated growth. However, this feeding method often leads to a range of issues, including digestive disorders, such as ruminal acidosis [3], reduced rumen microbial diversity and richness [4,5], impaired gut barrier function [6,7], abnormal serum biochemical parameters [8], and even oxidative stress [9], which ultimately decreases the growth performance and meat quality of fattening lambs [10].

The alkaline mineral complex (AMC) employs a molecular-scale microstructural design to construct a composite system of cell-activating agents based on aluminum compounds, silicon compounds, carbonate compounds, zinc oxide, and germanium dioxide. The preparation process integrates microbial fermentation engineering, molecular immunology, and animal nutrition regulation technology, leading to the formation of a stable ionic buffer system through microbial metabolic activity under strict anaerobic conditions [11]. AMC can promote the release of feed nutrients via physical wall-breaking effects, improve the feed conversion ratio, and enhance the animal’s antioxidant capacity, such as by increasing the levels of antioxidant-related indicators and their corresponding mRNA expression [12,13]. Additionally, AMC can effectively modulate the rumen environment by improving pH and regulating the bacterial population [14]. Furthermore, AMC produces beneficial effects by increasing the duodenal villus surface area and enhancing nutrient absorption [15]. While most existing studies on AMC have focused on their application in pigs [16], chickens [17], and cattle—particularly under conditions of weaning, transport, or heat stress [18]—their effect as a dietary additive in fattening lambs remains poorly understood. Therefore, based on the mechanisms outlined above, we hypothesized that supplementing AMC in lamb diets would improve the growth performance and meat quality of fattening lambs by buffering the ruminal pH, enhancing the systemic antioxidant capacity, and promoting intestinal nutrient absorption. Building on this rationale, the present study aimed to investigate the effects of AMC supplementation on the growth performance, meat quality, serum biochemical parameters, and digestive function of fattening lambs. The findings are expected to provide practical insights for the use of AMC as a nutritional strategy to enhance productivity in lamb production systems.

## 2. Materials and Methods

Ethical approval for all experimental procedures was obtained from the Animal Ethics Committee of Hebei Agricultural University (Approval No. 2020071).

### 2.1. Experiment Design

In this study, 96 Small-Tailed Han male lambs, weighing 48 ± 3.85 kg at six months of age, were weaned, vaccinated, and dewormed prior to the study. Lambs showing any signs of illness or extreme deviations in body weight were excluded. The lambs were divided into four treatment groups using a completely randomized design—the control group (CON, 0 g/d per lamb of AMC), test group 1 (LAMC, 2 g/d per lamb of AMC), test group 2 (MAMC, 3 g/d per lamb of AMC), and test group 3 (HAMC, 4 g/d per lamb of AMC)—with 3 pens in each group and 8 lambs per pen. All raw materials for AMC were purchased from Nail Biotechnology Co., Ltd., Beijing, China. The detailed composition and content of the additives are provided in Table 1. The concentrations for the treatment groups were determined based on a preliminary trial. The feeding trial lasted for 45 days. The trial animals were fed daily at 07:00 and 17:00 during the experimental period, with free access to feed and water. AMC was first thoroughly incorporated with Premix to ensure a homogeneous distribution. This mixture was then blended into the complete diet and processed into pellets, thereby ensuring complete intake by the lambs. The ingredients and nutritional composition of the basal diet are shown in Table 2.

### 2.2. Growth Performance

The feed offered and refused was recorded daily throughout the trial to measure dry matter intake (DMI). Body weight was recorded at the beginning and end of the experiment. The average daily gain (ADG) was calculated according to initial body weight (IBW) and final body weight (FBW). Feed conversion ratio (FCR) was calculated as DMI/ADG.

### 2.3. Carcass Traits and Meat Quality

At the end of the experimental period, to avoid pre-slaughter transport stress, 24 lambs (6 per group, with 2 lambs per pen) were randomly selected after an 8 h fast (with free access to water) and humanely slaughtered on-site at the experimental facility (Baoding Zhenhong Food Co., Ltd., Tang County, Baoding, China) for the determination of carcass traits and meat quality.

Carcasses were trimmed and weighed to determine the dressing percentage. The net meat weight was the weight of the carcass after the complete removal of bones, separable fat, and tendons. The Bone Weight was the weight of the bones after being separated from the meat. The saleable meat yield (SMY), sold to consumers in the form of retail cuts, was divided by the total weight of the carcass. The loin eye area was recorded on the cut surface of longissimus dorsi (LD) at the interface of the 12th and 13th rib on both sides of the carcass. The GR value (the depth of muscle and fat tissue from the surface of the carcass to the lateral surface of the 12th rib, 110 mm from the midline) was directly measured using a digital vernier caliper.

The meat samples used for meat quality assessment were stored at 4 °C for 24 h, after which the meat color, pH_24h_, marbling score, water-holding capacity (WHC), cooking loss, and shear force were measured, with the exception of pH_45min_. Each sample measurement was repeated three times. The LD was exposed by making a transverse cut between the 12th and 13th ribs. The marbling degree on the exposed cut surface was visually assessed by three trained panelists. The evaluation was conducted under consistent lighting conditions by comparing the muscle surface to standardized marbling reference photographs or charts. The final score for each carcass was the average of the panelists’ independent scores. The pH of the meat samples was measured at 45 min and 24 h after slaughter using a digital pH meter (S220 K, Mettler Toledo, Zurich, Switzerland). The meat color parameters were measured after 30 min of blooming at room temperature on the cut surface of the muscle (three different positions) using a CR-400 colorimeter (Konica Minolta, Tokyo, Japan). To measure the WHC, a circular meat sample weighing approximately 5 g was analyzed. Initially, the sample was placed on a platform with 18 layers of qualitative filter paper above and below and then subjected to a pressure of 35 kg using an RH-1000 meat pressure meter (Runhu Instrument Co., Ltd., Guangzhou, China) for 5 min. Finally, the WHC was determined as the difference between the initial and final weights of the sample and was expressed as a ratio relative to the original weight. For cooking loss measurement, 30 g of lamb meat was collected in a poly bag and placed in an 80 °C water bath until the core temperature reached 70 °C. Then, the lamb meat was removed from the water bath and cooled to room temperature, and we absorbed its moisture with white tissue paper. Weight loss of the sample was measured by deducting the moisture loss during the cooking process. The cooled meat samples were trimmed into regular rectangular blocks. Cylindrical cores (with a diameter of 1.27 cm) were then drilled along the muscle fiber direction, with at least three cores obtained per sample. Visible connective tissue and fat were meticulously removed. Shear force measurement was performed using a C-LM3 muscle tenderness meter (Beijing Bulader Technology Development Co., Ltd., Beijing, China), ensuring the blade traveled perpendicular to the direction of the muscle fibers. The maximum peak force required during the shearing process was recorded. The final shear force value for each sample was calculated as the average of the three measurements. A 5 g sample of clean muscle tissue, with visible fat removed, was placed in a cryotube and stored at −80 °C for subsequent determination of intramuscular fat (IMF) via the Soxhlet extraction method.

### 2.4. Serum Biochemical Parameters

Blood was collected from 6 lambs per group by jugular vein puncture at the end of the trial for serum biochemical analysis. Serum levels of alanine aminotransferase (ALT), aspartate aminotransferase (AST), lactate dehydrogenase (LDH), high-density lipoprotein (HDL), low-density lipoprotein (LDL), glucose (Glu), total protein (TP), blood urea nitrogen (BUN), total cholesterol (TC), triglyceride (TG), immunoglobulin A (IgA), immunoglobulin G (IgG), and immunoglobulin M (IgM) were determined using a fully automated biochemical analyzer (CLS880, Jiangsu Zecheng Biotechnology Co., Ltd., Beijing, China). Serum superoxide dismutase (SOD, Kit No.YX-191504C), glutathione peroxidase (GPx, Kit No.YX-071624C), total antioxidant capacity (T-AOC, Kit No.YX-200118C), malondialdehyde (MDA, Kit No.YX-130401C), and catalase (CAT, Kit No.YX-030120C) concentrations (all kits from BoRuiChangYuan Technology Co., Ltd., Beijing, China) were measured using a Thermo Multiskan Ascent fully automated microplate reader (Waltham, MA, USA).

### 2.5. Intestinal Morphology and Digestive Enzyme Content

After slaughtering, intestinal samples were taken from standardized anatomical sites: the duodenum (near the entrance of bile/pancreatic ducts), the jejunum (within the coiled mesenteric portion, away from flexures), and the terminal ileum (proximal to the ileocecal junction, identifiable by Peyer’s patches). From each site, a 2 cm segment was cut, rinsed gently with phosphate-buffered saline (PBS), and fixed with 4% paraformaldehyde for preservation. The samples fixed with 4% paraformaldehyde were removed, embedded in paraffin, sliced (thickness 5 mm), and stained with hematoxylin-eosin (HE). The villus height (VH), crypt depth (CD), and muscle layer thickness (MLT) were measured using a microscope (Motic BA210, Xiamen, China), and the ratio of villus height to crypt depth (V/C) was calculated. An additional 2 g of mucosal tissue from the duodenum, jejunum, and ileum was collected into 2 mL cryovials and stored at −80 °C for subsequent determination of lipase, protease, and amylase levels. The concentrations of digestive enzymes were determined using commercial enzyme-linked immunosorbent assay (ELISA) kits from Beijing Solarbio Science & Technology Co., Ltd. (Beijing, China), according to the manufacturer’s protocols.

### 2.6. Ruminal Fermentation Parameters and Microbial Composition

Rumen content (20 mL) was assessed in the slaughtered lambs. The rumen fluid was filtered through four layers of gauze into a beaker, and the rumen pH was immediately measured. The average value of three replicate measurements was calculated. The remaining filtered rumen fluid was stored at −80 °C until analysis of volatile fatty acids (VFAs) and ammonia nitrogen (NH_3_-N). Rumen VFA and NH_3_-N concentrations were analyzed using a QP2010 SE gas chromatograph (Shenzhen Ruisheng Technology Co., Ltd., Shenzhen, China). and the phenol–sodium hypochlorite colorimetric method, respectively.

The microbiota in the rumen samples were analyzed using metataxonomics. Briefly, microbial genomic DNA was extracted from rumen fluid samples using a specialized Stool DNA kit (Tiangen, Beijing, China). The purity and integrity of the genomic DNA were validated through 1.0% agarose gel electrophoresis. The V3-V4 regions of the bacterial 16S rDNA gene, spanning nucleotides 341 to 806, were amplified from the DNA using specifically designed barcoded primers: 341F (5′-CCTACGGGNGGCWGCAG-3′) and 806 R (5′-GGACTACNVGGGTATCTAAT-3′). Using a high-precision Biometra TOne 96 G PCR thermocycler (Jena, Germany), a PCR was carried out. The 50 μL reaction system incorporated 1.5 μL of each primer, 100 ng of template DNA, 5 μL of 10× KOD Buffer, 5 μL of 2.5 mM dNTPs, and 1 μL of KOD polymerase. The purified amplicons were combined in equimolar proportions and sequenced using paired-end sequencing on the Illumina HiSeq PE250 platform. Then, the sequences were clustered into operational taxonomic units (OTUs) at 97% sequence similarity. Alpha diversity metrics were calculated, including richness estimates (ACE and Chao1 indexes) and diversity indexes (Shannon and Simpson indexes). Principal coordinates analysis (PCoA) was used to compare the overall rumen microbiota based on unweighted UniFrac distance matrices. Disparities in taxonomic composition among samples were discerned using linear discriminant analysis (LDA) and effect size (LefSe) analysis.

### 2.7. Statistical Analysis

Data were collated in Excel 2010, and a single-factor analysis of variance (ANOVA) was performed using SPSS 29.0. Data are presented as the mean ± standard error of the mean (SEM). For variables where the ANOVA indicated a significant effect (*p* < 0.05), Duncan’s multiple range test was used as a post hoc measure to compare differences between group means. Significant differences are indicated in the tables.

## 3. Results

### 3.1. Growth Performance

No significant differences were observed in the IBW, FBW, and DMI between the groups (*p* > 0.05). The ADG of the MAMC group was significantly greater than that of the CON and HAMC groups, with a 14.41% increase over the CON group. The FCR of the MAMC group was significantly lower than that of the CON and HAMC groups (Table 3).

### 3.2. Carcass Traits and Meat Quality

There were no significant differences in regard to the dressing percentage, SMY, meat–bone ratio, loin muscle area, GR value, pH_45min_, pH_24h_, lightness, yellowness, marbling score, WHC, cooking loss, and shear force (*p* > 0.05). However, significant effects on redness (a*) and IMF content were observed. Specifically, the AMC-treated groups exhibited significantly higher redness (a*) values compared to the CON group (*p* < 0.05). Moreover, the IMF content of the MAMC group was significantly higher than that of the CON and HAMC groups (*p* < 0.05), with increases of 61.79% and 34.46% (*p* < 0.05), respectively (Table 4).

### 3.3. Serum Biochemical Parameters

The serum concentrations of LDL and TC in the HAMC group were significantly higher than those in the other groups (*p* < 0.01). Additionally, the HAMC group exhibited a significantly greater TG concentration compared to the LAMC group (*p* < 0.05) (Table 5).

There were no significant differences in the concentrations of IgG, IgA, and IgM between the groups (*p* > 0.05) (Table 6).

The serum SOD level in the AMC-treated groups was higher than that in the CON group (*p* < 0.05), and the serum MDA level in the HAMC group was higher compared to the MAMC group (*p* < 0.05). There were no differences in the serum GPx, CAT, and T-AOC among the groups (*p* > 0.05) (Figure 1).

### 3.4. Intestinal Morphology

The results of the HE staining (Figure 2) and measurements (Table 7) indicated that AMC supplementation had no apparent effect on the intestinal morphology of the fattening lambs, including the villus height, crypt depth, villus height to crypt depth ratio, and muscle layer thickness (*p* > 0.05).

### 3.5. Intestinal Enzyme Content

As shown in Table 8, compared with the CON group, the addition of AMC to the lambs’ diet produced no significant effect on the enzyme content in the ileum. The duodenum lipase content in the CON and MAMC groups was greater than that in the HAMC group (*p* < 0.05), and the MAMC group displayed significantly higher values than the LAMC group. The duodenal amylase concentration was significantly higher in the MAMC group (*p* < 0.05), with increases of 5.80% and 4.12% compared to the CON and LAMC groups, respectively. The HAMC group demonstrated a jejunal lipase concentration that was 28.71% lower than the average of the other groups. The jejunal amylase concentration was highest in the LAMC group among all groups (*p* = 0.060). Furthermore, the ileum amylase level increased with the AMC content (*p* = 0.053).

### 3.6. Rumen Fermentation Parameters and Microbial Composition

The AMC supplementation had no significant effect on the rumen fermentation parameters of fattening lambs (*p* > 0.05) (Table 9).

As shown in Figure 3, the number of OTUs unique to the four groups was 2372, 2511, 3000, and 2542, and there were 284 OTUs in the four sample groups. The AMC-treated groups had a higher Chao1 index and ACE index compared with the CON (*p* > 0.05). The Simpson and Shannon indices were relatively similar across the different groups (Figure 4).

The PCoA of the Bray–Curtis distance matrices (Figure 5) demonstrated that the AMC supplementation did not significantly alter the microbial composition and abundance in the rumen (*p* > 0.05), with no clear separation observed between the AMC-treated groups and the control group. The community compositions of the ruminal bacteria, as well as the top 10 phyla and genera with differential abundances, are illustrated in Figure 6. Bacillota, Bacteroidota, Proteobacteria, and Actinomycetota were the dominant phyla in the lambs’ rumen microbiota, accounting for over 90% of the total. Bacillota and Bacteroidota were absolutely predominant. There were no obvious differences in the relative abundances of the dominant phyla among the groups (*p* > 0.05), but the MAMC group exhibited the highest abundance of Bacillota. *Prevotella*, *Succinivibrionaceae_UCG*, *Dialister*, and *Lachnospiraceae_NK3A20_group* were the dominant genera in the lambs’ rumen microbiota, comprising more than 70% of the total. No significant differences were observed in the relative abundances of the dominant genera among the groups (*p* > 0.05), but the MAMC group exhibited the highest abundance of *Prevotella*.

To further clarify the effect of AMC on ruminal microorganisms, a LEfSe analysis of the MAMC group and the CON group (LDA > 2.0) was conducted. The results showed that the MAMC group significantly increased the levels of c_Synergistia, *g_Prevotellaceae_Ga6A1_group*, *g_Synergistes*, *g_Asteroleplasma*, and *g__Erysipelotrichaceae_UCG_009*. (Figure 6C,D).

### 3.7. Correlations Between Antioxidant Capacity, Growth, and Meat Quality

The correlation analysis (Figure 7) revealed that the serum GPx and T-AOC showed no significant correlations with the redness (a*) of the LD or ADG of fattening lambs. SOD and CAT were both positively correlated with meat redness (a*) but were not significantly associated with ADG. In contrast, MDA was negatively correlated with the ADG, while no significant relationship was observed for meat redness (a*).

## 4. Discussion

Growth performance is an important trait and is used as an indicator of animal production efficiency [19]. Previous studies have shown that AMC supplementation improves growth performance in a variety of animal models, including piglets [13], olive flounder [20], and calves [12]. In agreement with these studies, our results also indicated that supplementation with AMC increased lambs’ growth performance, as evidenced by the increased ADG and decreased FCR levels. Notably, the improved performance of lambs may be associated with an enhanced antioxidant capacity and the resultant improvement in their overall health. However, when the supplementation level was increased to 4 g/day/per lamb, no further improvement in growth performance was observed, implying that the lambs may enter a suboptimal metabolic state. Specifically, while the production performance of the group treated with the 4 g/d dose was no longer optimal (lower than that of the 3 g/d group), its results remained statistically comparable to those of the CON group. The mechanisms underlying this plateau effect merit further exploration.

Carcass traits are a key indicator of the meat production capacity [21]. Meat quality is a critical parameter that determines the palatability and market value of meat; the meat’s color and tenderness serve as important quality characteristics and key sensory attributes that consumers consider when making purchasing decisions [22]. The visual freshness and quality of lamb meat are signaled by its natural red color [23], mediated through the pigment myoglobin (Mb) [24], whose oxidation state in turn directly dictates color stability [25]. The IMF content is a key factor that influences tenderness. In our study, the AMC-treated group demonstrated a significantly higher redness level (a*) and an increased IMF content compared with the CON group. This is consistent with previous reports [26,27]. The enhancement in meat color stability observed in the present study may be linked to an improved systemic antioxidant status. We observed that the AMC supplementation in the MAMC groups significantly elevated the serum SOD level, a key enzyme for mitigating oxidative stress. Furthermore, our correlation analysis indicated a significant positive correlation between the serum SOD level and meat redness (a*), suggesting a direct role of the antioxidant status in preserving the meat color.

Serum biochemical parameters provide valuable insights into the underlying health status and metabolic abnormalities in animals [28]. Compared with the CON group, the SOD level was significantly increased in the AMC-treated groups. This result is consistent with the findings of previous research [29]. Although trace elements such as Mn, Cu, and Zn are known to be essential cofactors for antioxidant enzymes [30,31], and the basal diet in this study already met the nutritional requirements of the lambs, significant differences in antioxidant capacity were still observed among the groups. These differences may be attributed to the effects of other active components in AMC. AMC contains a relatively high level of silicon. Studies have shown that silicon (in the form of sodium metasilicate) exerts an antioxidant effect in murine RAW 264.7 macrophages by downregulating iNOS expression and reducing NO production [32]. In addition, low doses of silicon dioxide nanoparticles have also been shown to enhance the animal’s antioxidant capacity [33]. Secondly, AMC contains Na^+^, K^+^, and HCO_3_^−^, which are crucial for maintaining internal homeostasis. They can enhance the buffering capacity of the blood [34]. A stable and optimal acid–base environment is fundamental for SOD to perform its catalytic function at its best. Furthermore, the antioxidant defense system is a multi-layered and orderly relay system. SOD serves as the first line of antioxidant defense [31], specifically responsible for scavenging the initiating free radical—superoxide anion (O_2_^•−^)—and converting it into hydrogen peroxide (H_2_O_2_). The resulting H_2_O_2_ is subsequently processed by other enzymes such as CAT and GPx [35]. Under the experimental conditions of this study, we suggest that AMC intervention may have preferentially strengthened this primary line of defense. This could explain why the activities of other antioxidant enzymes did not increase simultaneously. The MDA concentration, as an indicator of oxidative stress, reflects lipid peroxidation [36]. In our study, the significant negative correlation between MDA and ADG, together with the observation that the HAMC group exhibited higher MDA concentrations and a weaker growth performance compared to the MAMC group, corroborates the adverse effect of oxidative stress on production performance. TC, LDL, and TG represent core lipid parameters. Atherosclerosis is initiated by the subendothelial retention of LDL [37], an accumulation whose extent is directly proportional to the levels of circulating LDL cholesterol, a primary component of TC. In the present study, the HAMC group exhibited significantly elevated serum TC and LDL concentrations, concomitant with reduced ADG and feed efficiency. These findings suggest that excessive AMC supplementation may disrupt lipid metabolic homeostasis, thereby impairing the overall production performance in animals. Dietary supplementation with AMC did not significantly alter the serum concentrations of immunoglobulins compared with the CON group. These results suggest that AMC, at the dosage used in this study, maintained the normal immune homeostasis of the animals.

The intestine is the main organ for digestion and absorption in animals, and its development and health are closely linked to growth performance [38]. The greater the height of the villi in the intestine, the greater the nutrient absorption [39]. However, in the present study, AMC supplementation did not significantly alter the intestinal morphology of the lambs. Nevertheless, AMC produced certain regulatory effects on the digestive enzyme content. Compared with the CON group, the amylase content in the duodenum was significantly increased in the MAMC group, which suggested that AMC may enhance carbohydrate hydrolysis by promoting amylase secretion, providing more available energy to the host. It is noteworthy that the lipase activity in both the duodenum and jejunum was significantly lower in the HAMC group than in the CON group, indicating a suppressive effect of the high-dose AMC on lipolytic function. Notably, the effects of AMC on intestinal digestive enzymes have rarely been reported, and the underlying regulatory mechanisms remain unclear. Further investigations are therefore warranted to elucidate how AMC modulates enzyme activities and the potential physiological implications. The integrity of the intestinal structure is primarily influenced by developmental stages, whereas digestive enzyme activities are subject to continuous fluctuations throughout the growth cycle, particularly during dietary transition phases [40]. This may explain why AMC supplementation modulated the digestive enzyme content without significant changes in the intestinal architecture in the current trial.

The rumen microbial community in ruminants constitutes a dynamic and complex ecosystem that not only facilitates the breakdown of feed and nutrient absorption but also plays a crucial role in maintaining the stability of the internal rumen environment [41]. The pH, NH_3_-N, and VFAs are key indicators of rumen microbial fermentation [42]. The level of NH_3_-N is closely related to the level of bacterial protein and serves as an important nitrogen source for microbial protein synthesis in the rumen [43]. Ruminal fermentation produces volatile fatty acids, which are the primary energy source for ruminants [44].

In our study, the AMC supplementation had no significant impact on ruminal fermentation parameters, indicating that AMC remains relatively stable in the rumen environment. However, changes in the ruminal microbial composition were observed. The MAMC group exhibited the highest number of OTUs, and the MAMC group displayed significantly increased *g_Prevotellaceae_Ga6A1_group* content according to the LEfSe analysis. This result aligns with previous findings reported by Liu et al. [45] and Cao et al. [46]. As a widespread and abundant genus in mammals, *Prevotella* is known to play a key role in fiber degradation, producing metabolites such as acetate and succinate, whose abundance is closely related to the host body, glucose metabolism, and intestinal health [47]. Several studies have demonstrated a strong correlation between the abundance of *Prevotella* and improved livestock growth performance [48,49,50]. Additionally, previous studies reported that *Prevotella* promotes fat deposition by altering the composition and content of fatty acids in IMF [51,52]. The latest research on Jinhua pigs has revealed that *Prevotella*, under high-fiber dietary conditions, modulates gut microbial communities and promotes host fat deposition, which was further validated in mouse models [53]. Importantly, that study provides novel mechanistic evidence supporting a direct causal link between the *Prevotella* genus and IMF deposition in animals.

Collectively, these results suggest that AMC supplementation can promote a beneficial shift in the rumen microbiome, potentially contributing to the improvements in growth performance and meat quality observed in this study through enhanced microbial ecological function (Figure 6).

Several limitations of this study should be acknowledged, which also point to clear directions for future research. First, when the supplementation level was increased to 4 g/day per lamb, no further improvement in growth performance was observed, suggesting that the lambs may have reached a metabolic plateau. The specific mechanisms remain to be elucidated. Furthermore, the influence of AMC on intestinal digestive enzymes and the associated regulatory pathways are currently unclear. Therefore, future investigations should prioritize elucidating the metabolic basis at higher doses and delineating the precise pathways through which AMC modulates digestive enzyme activity and their physiological implications.

To summarize our work, we have integrated our results into a speculative model for the AMC’s regulatory mechanism in fattening lambs (Figure 8).

## 5. Conclusions

Appropriate AMC supplementation can enhance growth performance and improve meat quality in fattening lambs by improving the antioxidant capacity and modulating the composition of beneficial rumen microbiota. Therefore, we recommend supplementing AMC at a dosage of 3 g/day/per lamb to optimize growth performance and meat quality, while avoiding potential adverse effects associated with excessive inclusion.

## Figures and Tables

**Figure 1 animals-16-00106-f001:**
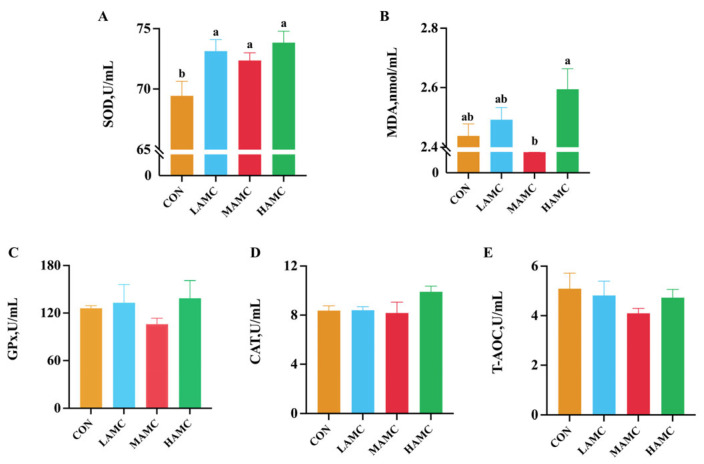
The effect of AMC supplementation on the antioxidant parameters of fattening lambs. (**A**) SOD = superoxide dismutase; (**B**) MDA = malondialdehyde; (**C**) GPx = glutathione peroxidase; (**D**) CAT = catalase; (**E**) T-AOC = total antioxidant capacity. ^a,b^ Values within a row with different superscripts differ significantly at *p* < 0.05.

**Figure 2 animals-16-00106-f002:**
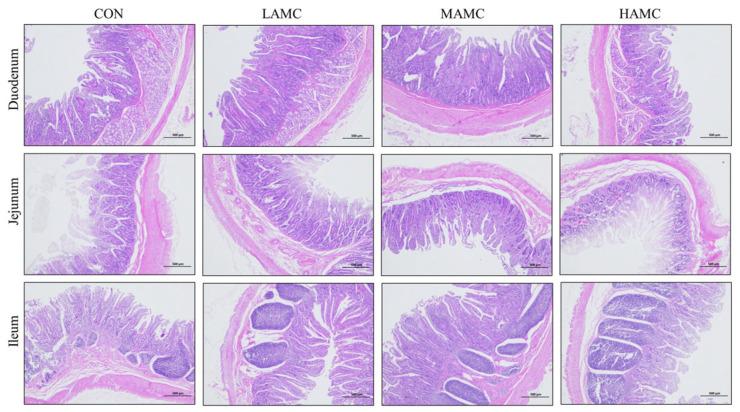
The effect of AMC supplementation on the small intestine morphology, illustrated by HE staining.

**Figure 3 animals-16-00106-f003:**
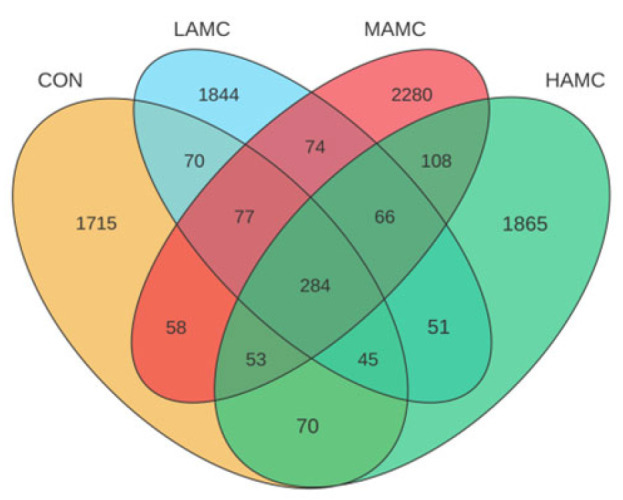
Venn diagram illustrating unique and common OTUs among four groups. OTUs = operational taxonomic units.

**Figure 4 animals-16-00106-f004:**
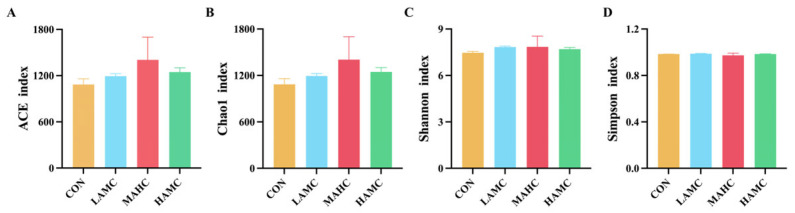
Alpha diversity analysis of rumen flora: (**A**) ACE index of species richness; (**B**) Chao1 index of species richness; (**C**) Shannon index of species diversity; and (**D**) Simpson index of species diversity.

**Figure 5 animals-16-00106-f005:**
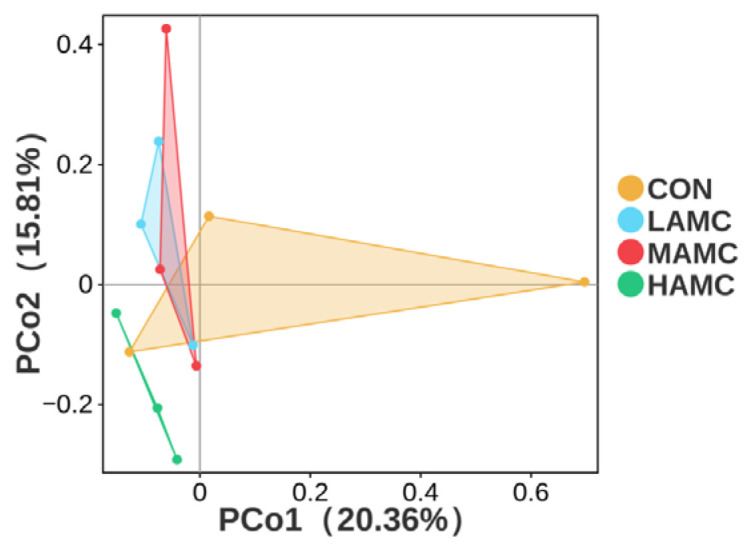
PCoA plot based on unweighted UniFrac distance matrices comparing the rumen bacterial communities. PCoA = principal coordinates analysis.

**Figure 6 animals-16-00106-f006:**
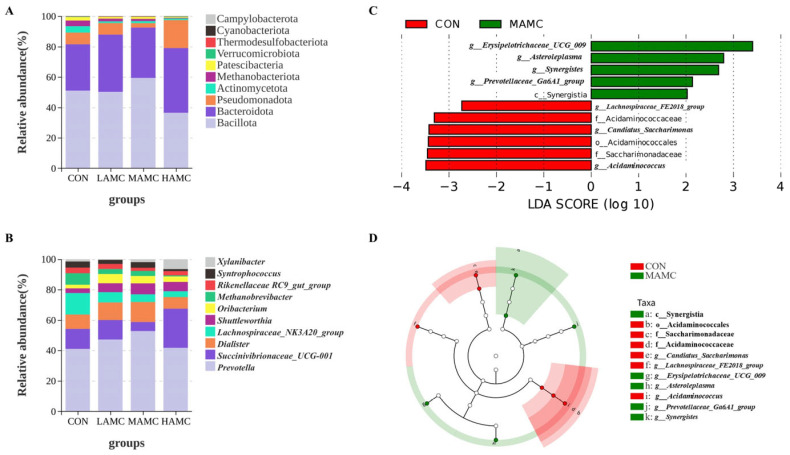
Ruminal bacterial compositions in the four groups: (**A**) Relative abundance of phyla. (**B**) Relative abundance of genera. (**C**) LDA score plot of differentially abundant microbial taxa between the CON and MAMC groups. (**D**) LEfSe cladogram showing differential abundance of microbial taxa between the CON and MAMC group. LDA = linear discriminant analysis.

**Figure 7 animals-16-00106-f007:**
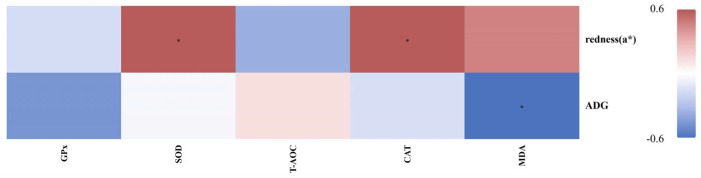
Correlation analysis of antioxidant capacity for growth performance and meat quality attributes in fattening lamb. * indicates a statistically significant correlation at *p* < 0.05.

**Figure 8 animals-16-00106-f008:**
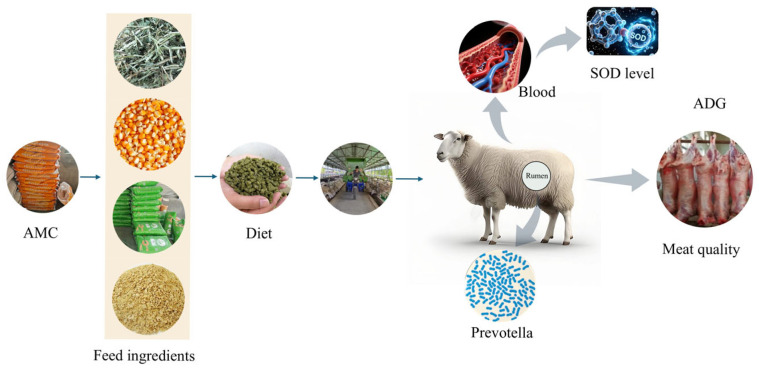
The regulatory mechanism of AMC on the fattening of lambs that we have speculated and summarized.

**Table 1 animals-16-00106-t001:** The calculated ion content of AMC supplementation.

Ions	Calculated Contents, mg/kg
SiO_3_^2−^	49,473.00
Na^+^	29,943.24
K^+^	26,910.00
Zn^2+^	5.52
Ge^4+^	0.14
HCO_3_^−^	42,090.00

**Table 2 animals-16-00106-t002:** Ingredients and nutritional composition of the basal diet (air-dried basis).

Ingredients	Percentage (%)
Peanut hay	15.61
Soybean meal	15.94
Linseed meal	8.94
Corn	54.58
Premix ^1^	3.35
Yeast	0.11
NaHCO_3_	1.26
NaCl	0.21
Nutritional composition ^2^	
Crude protein	18.50
Crude fat	3.34
Acid detergent fiber	12.04
Neutral detergent fiber	20.25
Ca	0.75
P	0.60
Metabolic energy, MJ/kg	12.62

^1^ The premix provided the following nutrients per kilogram: vitamin A 200,000 IU, vitamin D3 35,000 IU, vitamin E 500 mg, Zn 800 mg, Cu 90 mg, Mn 900 mg, I 15 mg, Se 5 mg, calcium hydrophosphate 20 g, and NaCl 10 g. ^2^ Nutritional composition includes measured values.

**Table 3 animals-16-00106-t003:** The effect of AMC supplementation on the growth performance of fattening lambs.

Items	CON	LAMC	MAMC	HAMC	SEM	*p*-Value
IBW, kg	48.00	48.43	48.22	48.20	0.415	0.989
FBW, kg	61.74	62.38	63.61	61.23	0.535	0.411
ADG, g/d	292.24 ^b^	296.86 ^ab^	334.35 ^a^	277.13 ^b^	7.200	0.029
DMI, g/d	1757.83	1626.05	1802.64	1675.87	0.039	0.425
FCR	6.01 ^a^	5.48 ^ab^	5.40 ^b^	6.05 ^a^	0.115	0.049

IBW = initial body weight; FBW = final body weight; ADG = average daily gain; DMI = dry matter intake; and FCR = feed conversion ratio. ^a,b^ Values within a row with different superscripts differ significantly at *p* < 0.05.

**Table 4 animals-16-00106-t004:** The effect of the AMC supplementation on the carcass traits and meat quality of fattening lambs.

Items	CON	LAMC	MAMC	HAMC	SEM	*p*-Value
Dressing percentage, %	52.57	52.82	52.44	53.09	0.002	0.674
SMY, %	68.82	69.09	68.89	68.93	0.004	0.984
Meat–bone ratio	3.06	3.07	3.07	3.07	0.055	1.000
Loin muscle area, cm^2^	49.10	53.05	52.42	52.10	1.310	0.745
GR value, mm	26.99	25.18	27.51	24.88	0.938	0.729
pH_45min_	7.39	7.52	7.18	7.03	0.071	0.053
pH_24h_	5.82	5.78	5.82	5.80	0.020	0.777
Meat color						
a*	13.38 ^b^	15.79 ^a^	15.87 ^a^	17.68 ^a^	0.475	0.010
L*	38.13	40.17	39.81	39.70	0.430	0.106
b*	8.31	9.01	9.12	9.62	0.280	0.444
Marbling score	2.50	2.33	3.00	2.33	0.134	0.255
WHC, %	50.83	53.33	51.17	50.67	0.005	0.292
Cooking loss, %	34.11	36.15	33.33	31.33	0.011	0.525
Shear force, N	41.12	38.98	34.71	35.47	1.421	0.361
IMF, %	2.46 ^b^	3.32 ^ab^	3.98 ^a^	2.96 ^b^	0.184	0.013

SMY = saleable meat yield; WHC = water-holding capacity; and IMF = intramuscular fat. ^a,b^ Values within a row with different superscripts differ significantly at *p* < 0.05.

**Table 5 animals-16-00106-t005:** The effect of AMC supplementation on the serum biochemical contents of fattening lambs.

Items	CON	LAMC	MAMC	HAMC	SEM	*p*-Value
ALT, U/L	21.85	23.71	22.44	25.20	1.436	0.751
AST, U/L	105.87	124.28	114.38	121.74	2.794	0.200
LDH, U/L	503.10	558.14	573.08	611.50	17.667	0.181
HDL, mmol/L	0.81	0.77	0.77	0.90	0.029	0.194
LDL, mmol/L	0.72 ^b^	0.59 ^b^	0.65 ^b^	0.98 ^a^	0.052	0.001
Glu, mmol/L	4.27	4.22	4.28	4.49	0.124	0.806
TP, g/L	65.63	65.60	65.33	66.88	0.589	0.618
BUN, mmol/L	8.39	7.42	7.42	7.65	0.280	0.766
TC, mmol/L	1.71 ^b^	1.54 ^b^	1.61 ^b^	2.16 ^a^	0.082	0.001
TG, mmol/L	0.26 ^ab^	0.22 ^b^	0.24 ^ab^	0.32 ^a^	0.017	0.020

ALT = alanine aminotransferase; AST = aspartate aminotransferase; LDH = lactate dehydrogenase; HDL = high-density lipoprotein; LDL = low-density lipoprotein; Glu = glucose; TP = total protein; BUN = blood urea nitrogen; TC = total cholesterol; and TG = triglyceride. ^a,b^ Values within a row with different superscripts differ significantly at *p* < 0.05.

**Table 6 animals-16-00106-t006:** The effect of AMC supplementation on the immune parameters of fattening lambs.

Items	CON	LAMC	MAMC	HAMC	SEM	*p*-Value
IgG, g/L	15.60	15.47	15.01	15.80	0.303	0.705
IgA, g/L	0.63	0.53	0.53	0.52	0.016	0.064
IgM, g/L	1.40	1.15	1.11	1.27	0.059	0.246

IgA = immunoglobulin A; IgG = immunoglobulin G; and IgM = immunoglobulin M.

**Table 7 animals-16-00106-t007:** The effect of AMC supplementation on the intestinal morphology of fattening lambs.

Items	CON	LAMC	MAMC	HAMC	SEM	*p*-Value
Duodenum						
Villus height, mm	0.52	0.67	0.46	0.51	0.038	0.274
Crypt depth, mm	0.46	0.53	0.42	0.46	0.027	0.735
V/C	1.16	1.68	1.22	1.13	0.172	0.666
Muscle layer thickness, mm	0.35	0.27	0.34	0.40	0.021	0.195
Jejunum						
Villus height, mm	0.54	0.48	0.48	0.45	0.027	0.706
Crypt depth, mm	0.42	0.35	0.43	0.40	0.023	0.652
V/C	1.41	1.44	1.17	1.20	0.105	0.763
Muscle layer thickness, mm	0.27	0.24	0.23	0.29	0.019	0.726
Ileum						
Villus height, mm	0.48	0.60	0.53	0.56	0.018	0.119
Crypt depth, mm	0.33	0.34	0.36	0.40	0.018	0.446
V/C	1.49	1.87	1.52	1.44	0.076	0.150
Muscle layer thickness, mm	0.27	0.31	0.31	0.30	0.019	0.153

V/C = villus height to crypt depth ratio.

**Table 8 animals-16-00106-t008:** The effect of AMC supplementation on the intestinal enzyme content of fattening lambs.

Items	CON	LAMC	MAMC	HAMC	SEM	*p*-Value
Duodenum						
Lipase, IU/L	416.14 ^ab^	370.71 ^bc^	447.68 ^a^	360.02 ^c^	10.830	0.004
Protease, IU/L	450.16	476.24	491.81	445.73	12.225	0.517
Amylase, IU/L	260.73 ^b^	264.92 ^b^	275.84 ^a^	270.00 ^ab^	1.901	0.027
Jejunum						
Lipase, IU/L	461.98 ^a^	468.51 ^a^	457.16 ^a^	329.76 ^b^	20.253	0.022
Protease, IU/L	390.62	402.4	371.62	311.97	19.015	0.359
Amylase, IU/L	275.16	298.98	275.60	262.25	4.732	0.060
Ileum						
Lipase, IU/L	349.11	315.91	333.58	330.33	10.271	0.788
Protease, IU/L	332.10	334.17	300.75	270.81	11.740	0.200
Amylase, IU/L	246.97	249.33	259.59	261.27	2.318	0.053

^a,b,c^ Values within a row with different superscripts differ significantly at *p* < 0.05.

**Table 9 animals-16-00106-t009:** The effect of AMC supplementation on the rumen fermentation parameters of fattening lambs.

Items	CON	LAMC	MAMC	HAMC	SEM	*p*-Value
pH	5.57	5.68	5.88	5.76	0.094	0.728
NH_3_-N, mg/dL	17.62	24.92	21.48	18.88	2.471	0.760
Acetate, mmol/kg	50.70	53.47	43.35	46.23	2.463	0.498
Propionate, mmol/kg	35.18	34.25	34.95	31.45	2.864	0.971
Butyrate, mmol/kg	15.68	15.70	11.48	16.55	1.042	0.323
Isobutyrate, mmol/kg	0.98	1.252	0.88	0.93	0.072	0.285
Valerate, mmol/kg	3.14	3.56	3.40	2.96	0.214	0.790
Isovalerate, mmol/kg	2.28	3.02	1.82	2.10	0.267	0.455
Caproate, mmol/kg	0.62	0.82	0.50	0.86	0.085	0.445

NH_3_-N = ammonia nitrogen.

## Data Availability

The data presented in this study are available upon request from the corresponding authors.

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
