# Peer review of "Effects of Alkaline Mineral Complex Supplementation on Growth Performance, Meat Quality, Serum Biochemical Parameters, and Digestive Function of Fattening Lambs"

_animals, 2025, doi:10.3390/ani16010106_

Round 1
Reviewer 1 Report
Comments and Suggestions for Authors
The research presented on AMC supplementation is novel and relevant, as it explores an additive and its effects in ruminants, evaluating both productive parameters and changes in the ruminal microbiota and intestinal tissue. This integrated approach provides valuable information based on the results obtained. Respectfully, I recommend that the authors consider the following observations to improve the manuscript. I have indicated in green-highlighted text some commented points to help identify certain observations

Reviewer 2 Report
Comments and Suggestions for Authors
This study has scientific merit and is suitable for publication in Animals. However, it needs corrections. All my suggestions are in the attached file.

This study has scientific merit and is suitable for publication in Animals. However, it needs corrections. All my suggestions are in the attached file.
Reviewer 3 Report
Comments and Suggestions for Authors
Dear authors, thank you for the well-written manuscript (MS) you provided and your interesting work. However, I have some crucial comments that need to be addressed in order your MS to be published.
Simple summary, Abstract and Introduction sections are clear of comments.
Materials and Methods
i) I would like to know why only male lambs were selected. Do the authors think that pubity is not linked to growth performance, as well as testosterone concentration?
ii) How was the selection and randomisation of lambs performed?
iii) Were the selected lambs from ewes with known high growth performance and etc?
iv) GR index needs to be briefly explained.
v) Why was extra supplementation with Zn selected?
vi) How was it ensured that every lamb was eating the exact amount of AMC supplemented ratio, and no lower or higher amounts?
vii) Paragraph 2.6: the reference 14 needs to be briefly explained, and some details about the indices used.
Results
i) Figure 1: Needs quality improvement.
ii) The paragraph referring to fig. 3 needs to be explained in Materials and Methods section
iii) Which are the Chao1 and ACE indices? Reference needs to be made in Materials and Methods section
iv) Line 239: in Fig 4 is of _ rumen (a "space" must be added)
v) "PCoA of Bray-Curtis distance matrices" needs to be explained in Materials and Methods section.
Discussion
i) Lines 294-297 can be combined in one sentence.
ii) Lines 323-324 imply that SOD activity is increased due to the abundance of metal cofactors. However, in order for the activity of SOD to be increased, it is known that a "trigger" related to the enzyme's production or involved in the antioxidant pathway and mechanism containing SOD needs to be activated. In this MS can the authors provide data that only the abundance of metal ions can lead to an increase in SOD activity, since no other antioxidant enzyme or parameter is significantly affected by the AMC supplementation?
iii) Lines 332-334 are speculation, and no possible biochemical pathway or reference is provided. Which biochemical pathway do the authors believe is affected by the AMC supplementation regarding amylase secretion?
iv) Lines 334-336: In HAMC group MDA levels are also elevated, which may explain the impairment in lipolytic function.
v) Lines 382-383: In MAMC group MDA levels are not significantly lower than control group, so how can it be implied that AMC supplementation lowers MDA levels?
vi) Lines 387-388: How can this assumption be explained?
vii) Line 389: MDA refers to lipid peroxidation as an indicator of oxidative stress.
viii) No limitations and future steps are suggested.
ix) The results of LDL, TC, and TG are not discussed, even though they are interesting findings.
As a conclusion, the discussion section needs references to be added in order to give more strength to your findings and arguments. The antioxidant argument needs more proof to be supported. Otherwise, it is considered an assumption.
Reviewer 4 Report
Comments and Suggestions for Authors
The manuscript addresses an interesting topic regarding the effects of AMC supplementation on lamb performance, rumen fermentation, and microbial composition. The study presents a substantial dataset and includes multiple physiological and microbiological parameters. However, the manuscript has several major issues that need to be addressed before it can be considered for publication.
My suggestions are follow:
L47, Please arrange the keywords in alphabetical order
L83, Please detail the randomization procedure (by initial BW or simple randomization). Clarify the experimental unit used in statistical analysis (individual lamb or replicate pen). Provide justification or power calculation for sample size.
L106, Describe selection method for slaughtered animals (24 lambs were slaughtered (6/treatment). Please explain how these 24 animals were randomly selected (whether 2 animals per replicate).), and transport distance/time, lairage/fasting duration, and measures to minimize pre-slaughter stress.
L120, blood collected from 6 lambs per group? Is the sample size adequate?
L131, Please describe the specific sampling location and how to define jejunum, ileum, etc?
L149, Please briefly describe the method for the determination of rumen bacteria.
L199, provide kit catalog numbers
L228, Regarding rumen NH3-N, the values reported for the LAMC and MAMC groups exceed 20 mg/dL, which is notably higher than the generally accepted physiological and functional ranges in ruminants
L285-310, This paragraph is informative but currently reads more like a review. The authors should reduce general background content and strengthen the interpretation of their own results, especially regarding the mechanisms by which AMC affects meat color and tenderness.
L311-324, The explanation of SOD function (superoxide conversion, metal cofactors, etc.) is too detailed for a discussion section. A concise summary would be more appropriate. And only SOD is discussed, even though many biomarkers were measured (ALT, AST, HDL, LDL, MDA, TP, IgG, etc.). The authors need briefly explain why other indicators showed no significant differences, or discuss any additional significant findings.
L404, Conclusion state that the 3g/d (MAMC group) performs best overall. However, one important parameter does not fully align with this interpretation. Total VFA is commonly used as an indicator of overall ruminal fermentation activity, and the MAMC group (96) shows the lowest concentration among all treatments, including the control (108).
The manuscript contains numerous grammatical errors and language issues that significantly affect readability. We recommend that the authors have the manuscript thoroughly reviewed and edited by a native English speaker or a professional language editing service. Some examples of recurring issues, though not exhaustive, include:
L272, “And research on...” change to “A previous research on .....” or “A previous study in ....”
L274, “dietary supplementation of Barodon” change to “ dietary supplementation with Barodon”
L275, “and drinking water AMC supplementation...” change to “and supplementation with AMC in drinking water improved...”
L351, “However, change in ruminal microbial composition were observed” change to “However, changes in the ruminal microbial composition were observed, indicating that AMC may influence microbial diversity despite minimal impact on fermentation parameters.”
Comments on the Quality of English LanguageA thorough professional language editing is also strongly advised.
Round 2
Reviewer 1 Report
Comments and Suggestions for Authors
Thank you for your careful revisions and for addressing the comments raised during the first round of review. The manuscript has improved notably in terms of clarity, structure, and consistency. The following are minor comments intended to further enhance precision, consistency of terminology, and clarity of the experimental description:
L10. Use “animals” instead of “lambs” to avoid repeating the word continuously in this sentence.
L15. Remove “the lambs” and make the corresponding grammatical correction. It is assumed that the weight gain refers to lambs
L30-L31. For consistency throughout the document, please use “lambs.”
L30. Is it possible that the authors are referring to the term “replicate” as the pens? There is a comment in L119 that could be considered to improve the clarity of this information.
L56-L58. The authors may consider that at least two references are required for the first paragraph of this Introduction section.
L100. Please clarify how the supplementation levels were established. Although this may seem obvious, the sentence does not clearly describe whether this refers to the AMC concentration in the diet. Additionally, please provide details of the preliminary trial, including its main characteristics.
L115. Please revise this section, as some abbreviations have already been defined previously and should not be repeated.
L119. The description “6 per treatment, with 2 lambs per pen” seems more appropriate for clearly defining the experimental units, including animals and pen as the experimental unit. This type of description may be better suited for sections such as the Abstract. In addition, consider identifying AMC levels as “treatments” rather than “groups,” and using “pen” instead of “replicate.” For example, the sentence “Each group contained 24 sheep, with three replicates per group and 8 sheep per replicate” could be revised accordingly to improve clarity.
Reviewer 2 Report
Comments and Suggestions for Authors
This study has scientific merit and is suitable for publication in Animals.
Comments on the Quality of English LanguageThis study has scientific merit and is suitable for publication in Animals.
Author Response
We sincerely thank the reviewer for their positive assessment of our work and for recommending it for publication.
Reviewer 3 Report
Comments and Suggestions for Authors
Dear authors,
The MS is revised but I still have some core questions that remained half-answered.
A general question why were the animals on 8-hour food deprivation before slaughtering?
Lines 80-81: REFs need to be added.
Line 368: Provide ref otherwise it is considered as speculation. Additionally, how did the 8 hours food deprivation before slaughtering affect the antioxidant capacity of the animals and the authors suggest that the animals were not stressed? In my knowledge removing the routine towards the feeding hours causes stress in animals, even the extra personnel.
In the slaughtered animals and only, were the levels of SOD measured before the 8-hour food deprivation and right before the slaughtering? If not please explain the reason.
Line 373: Add a REF otherwise is not actually relevant and accurate and shall be removed, because in the paragraph the authors discuss meat and in the certain sentence they refer to muscle from a living animal.
Lines 431-433: If one study why two refs then?
Regarding my previous comment " I would like to know why only male lambs were selected. Do the authors think that pubity is not linked to growth performance, as well as testosterone concentration?" the lambs aged 6 months are not sexually mature indeed, but the rising Testosterone levels exist, so why did the authors exclude the testosterone effect and as well as the individuality of animals as a parameter?
Regarding my comment "How was the selection and randomization of lambs performed?" your answer "A total of 96 healthy crossbred male lambs with an average initial body weight of 48 ± 3.85 kg and aged approximately 6 months were selected for the trial. All lambs were weaned, vaccinated, and dewormed prior to the study. Lambs showing any signs of illness or extreme deviations in body weight were excluded." must be incorporated in Materials and Methods section.
Regarding my comment "Why was extra supplementation with Zn selected?" I need further explanation and clarification. In the standard ratio Zn is added in 800mg so why extra Zn is needed? In case that Zn is in deficiency then the result of increased SOD activity can be explained by the Zinc addition and only. My question is, why to use AMC as an antioxidant enhancer when just the supplementation of Zn has probably the same result, since it is proved that solely Zn supplementation enhances SOD activity? Your results show that AMC is probably involved only in SOD activity with no other antioxidant effect. The MDA levels support the theory that AMC in 4g/d do more harm than good, and in lower dosage has no effect on MDA levels. So, no other implication in antioxidant variants is observed with AMC supplementation.
In overall, discussion section is improved, but not in a satisfying way regarding antioxidant- and enzyme-linked pathways.
Also, more references need to be added regarding every parameter discussed.
Finally, a future step is to elucidate the antioxidant and enzymatic pathways and it should be included in the discussion.
Reviewer 4 Report
Comments and Suggestions for Authors
The authors have made extensive revisions and provided detailed explanations in the response document. However, some of these responses have not been fully incorporated into the revised manuscript itself. For example, clarifications regarding the definitions of intestinal segments (e.g., jejunum and ileum) and the exact sampling locations are explained in the response file but are not clearly reflected in the Methods section of the manuscript.
As these details are essential for transparency and reproducibility, they should be explicitly included in the manuscript rather than only addressed in the supplementary response. This issue should be carefully checked during further revision.
In addition, the following suggestions are provided for the authors’ consideration:
L67. After spelling out the full term at its first mention, the abbreviation should be used consistently throughout the manuscript. Please check and revise the entire manuscript accordingly.
This sentence contains grammatical and logical inconsistencies
L284-285, HAMC vs. , the other groups, ....the next skip to LAMC,
L285, concentration of jejunum amylase change to jejunal amypase concentration.
L286, LAMC was highest in the CON and AMC..., “highest” meaning only one, rather than two groups (CON, AMC.)
L381, deleted “.”
L384, corroborates change to corroborate.
L349, that supplementation with AMC...(deleted “a”)...or AMC supplementation
Comments on the Quality of English LanguageLanguage quality has improved compared with the previous version, but some long sentences and awkward phrasing remain. A final round of professional language editing is still recommended.
